# The Skin and Gut Microbiome and Its Role in Common Dermatologic Conditions

**DOI:** 10.3390/microorganisms7110550

**Published:** 2019-11-11

**Authors:** Samantha R. Ellis, Mimi Nguyen, Alexandra R. Vaughn, Manisha Notay, Waqas A. Burney, Simran Sandhu, Raja K. Sivamani

**Affiliations:** 1PotozkinMD Skincare Center, Danville, CA 94526, USA; samantha.rose.ellis@gmail.com; 2Department of Dermatology, University of California-Davis, Sacramento, CA 95816, USA; missallievaughn@gmail.com (A.R.V.); mnotay@gmail.com (M.N.); waqas.ahmed.burney@gmail.com (W.A.B.); 3School of Medicine, University of California-Davis, Sacramento, CA 95817, USA; nguyen.mimi.ph@gmail.com (M.N.); youngsimbaa32@gmail.com (S.S.); 4Department of Biological Sciences, California State University, Sacramento, CA 95819, USA; 5College of Medicine, California Northstate University, Elk Grove, CA 95757, USA; 6Pacific Skin Institute, Sacramento, CA 95815, USA; 7Zen Dermatology, Sacramento, CA 95819, USA

**Keywords:** gut, skin, microbiome, acne, psoriasis, rosacea, atopic dermatitis

## Abstract

Microorganisms inhabit various areas of the body, including the gut and skin, and are important in maintaining homeostasis. Changes to the normal microflora due to genetic or environmental factors can contribute to the development of various disease states. In this review, we will discuss the relationship between the gut and skin microbiome and various dermatological diseases including acne, psoriasis, rosacea, and atopic dermatitis. In addition, we will discuss the impact of treatment on the microbiome and the role of probiotics.

## 1. The Skin Microbiome

Mary Marples’s article, “Life on the human skin”, parallels the diverse ecosystems of the skin to those of our Earth [1]. Skin is likened to soil, with the human host being the Earth on which these microcosms exist. Both environments support life and exert selective pressures on these organisms, from “… the desert of the forearm, [to] the tropical forest of the armpit, [and] the cool woods of the scalp”. The skin is much more than just its resident constituents—the skin is a rich ecosystem that supports diverse populations of organisms, and like all life, these microorganisms, too, are competing for their chance to survive. In this light, skin microbiome research can evolve beyond focusing on the analysis of skin inhabitants and instead foster a respect for the beautifully integrated portrait before us: the system and its ecology. By focusing on the interactions between the biological and ecological systems, one has the chance to appreciate this portrait of a unique environment and its equally unique residents coming together to form a dynamic ecosystem.

The skin is comprised of three major habitats: moist, sebaceous, and dry. Sebaceous skin includes the face, chest, and back, and is a comparatively simple community, composed mainly of several species of *Cutibacterium* (formerly *Propionibacterium*), *Staphylococcus* bacteria, and *Malassezia* yeasts [2,3,4]. Sebum excretion appears to be the primary driving force in sebaceous microbiome development and maintenance, as sebaceous microbiomes shift dramatically around puberty when oil production increases [3]. 

Dry skin sites, such as the arms and legs, are also dominated by *Cutibacterium acnes* and *Staphylococcus* species, but with additional significant portions of Gammaproteobacteria and Betaproteobacteria [5]. Moist sites have more variability, with the constantly moist toe web favoring *Corynebacterium* growth, while the phalangeal web, which features abundant sweat glands but is generally drier, favoring *Staphylococci* [5]. Invaginations of the skin (e.g., hair follicles, sebaceous glands, and sweat glands) create distinct microenvironments and oxygen gradients which may promote the growth and colonization of particular microbes. For example, *Cutibacteria* are oxygen tolerant anaerobes, though they grow much faster in truly anaerobic environments, while *Staphylococci* are facultative anaerobes and grow fastest in the presence of oxygen. It is important to note that most physiological research on *Staphylococcus* species has been done during aerobic growth [6,7,8,9,10,11], so the effects of hypoxic or anoxic conditions on metabolism is unknown.

It is apparent that the skin microbiome is important in homeostasis, partially through the maintenance of the cutaneous immune system. For example, some strains of *S. epidermis* have been found to enhance the innate barrier immunity and activate IL-17+ CD8 T cells to protect against infection [12]. In a study by Naik et al., mice raised in germ-free conditions exhibited a reduction in IL-17A production in the skin, that was reversible with subsequent *S. epidermis* colonization [13]. In addition, *S. epidermis* is found to induce CD8 T cell associated transcripts important in promoting tissue repair [14]. Although further research is needed to fully elucidate the workings of the cutaneous immune system, it is evident that residential microbes play an important role.

## 2. Gut: Local and Systemic Modulation

The human gastrointestinal tract is home to several different microbial ecosystems that colonize the entire mucosal lining [15,16]. This dynamic system is influenced by genetics, diet, and several other environmental factors [17]. Nearly 10 million genes have already been identified in the gut microbiome [18], many of which are used to support the human genome in performing several important and essential functions like vitamin production, immune regulation, protection from pathogens, serum lipid modulation, and metabolism of xenobiotics and food components [19,20,21]. The resulting metabolites may also influence metabolism within the host, demonstrating that both the human genome and gut microbiome play a role in the metabolic pathways occurring in the human body [22,23]. The catabolic end products from the fermentation of complex carbohydrates and other undigested food components by the intestinal microbes are incorporated into the body’s short chain fatty acids (SCFAs). Therefore, any change in the gut microbiota’s composition or metabolic activity may also alter fatty acid levels [23,24]. 

Fermentation of prebiotics by the gut microbiota can also produce SCFAs, which may improve the function and integrity of the gut, modulate the immune system and inflammatory response, and affect lipid and glucose metabolism [25]. In fact, these byproducts may be anti-tumorigenic, as SCFAs, butyrate, acetate, and propionate, produced by the fermentation of dietary fibers by colonic microbes, have also been shown to induce apoptosis in colorectal tumor cells [26]. There is emerging evidence that free fatty acids (FFAs), in addition to serving as an important sources of energy, are also involved in several biological processes including modulation of gene expression of adipocytes, macrophages, and endothelial cells [27,28,29,30]. 

FFAs can also modulate cytokine and chemokine production, gene expression of adhesion molecules, and have pro-resolution and anti-inflammatory properties, thereby controlling inflammation at multiple levels [18,28,29,31,32,33]. Javier et al. showed that increased intestinal colonization of *Akkermansia* was the major predictor of serum total FFA levels, and was negatively related to the total FFA and IL-6 (a proinflammatory cytokine) levels. He also found that altered serum levels of FFAs were associated with an imbalance between *Lactobacillus* and *Akkermansia*, as well as increased serum IL-6 levels, fecal SCFA, and subclinical prevalence of metabolic alterations [24].

Gut microbiota may also convert excess proteins and amino acids into certain toxins, like indoxyl sulfate, trimethylamine N-oxide (TMAO), and p-cresyl sulfate, which may be involved in a number of diseases [34]. Current evidence suggests that the most effective way to improve the microbiotic profile is increasing dietary fiber, which results in an increased synthesis of SCFAs by the gut microbiome and decreased levels of certain toxic molecules [35]. In addition, supplementation with omega-3 polyunsaturated fatty acids also increases SCFA-producing bacteria [36]. Taken together, the data highlights an important relationship between human gut microbiota, its metabolism, and its related effects on overall human health.

## 3. Role of Diversity in the Microbiome

The gut and skin microbiota are made up of trillions of microbes, derived from thousands of different strains, that live together as an intricate ecological community. These microbiota, along with their metabolic byproducts and host interactions, directly influence both normal physiology and disease processes. For instance, disruption of the normal symbiotic relationship between gut microbes and the host is associated with inflammatory bowel disease (IBD) [37,38], obesity [39], and metabolic syndrome [40]. We are still trying to understand how microbial diversity in the gut and on the skin influences health, but there is evidence that increased microbial diversity overall is associated with improved physiology and homeostasis [15].

It is important to note that diversity of microbes in the intestines varies drastically across human populations and cultures and even varies significantly within healthy individuals over time [41]. The microbiome changes considerably over the first three years after birth and is highly sensitive to environmental factors such as breastfeeding and antibiotics [42]. It is generally accepted that two phyla, *Bacteroidetes* and *Firmicutes*, usually predominate in the microbiome of adults, while *Actinobacteria* and *Proteobacteria* make up a smaller portion [15]. Even so, there can still be variations in the proportions of these phyla and in the species from person to person. 

Another important component that diversifies the microbiome is the diversity of functional gene profiles within the gut. Characterization of genome content through sequencing of cultured gut isolates offers important information about the functional capacity of the gut microbiome. Interestingly, one study showed that despite having very different microbial compositions in the gut, a group of 18 females shared over 93% of the same enzyme level functional groups [43]. This study demonstrated that diverse communities of bacterial phyla and strains nonetheless yield highly similar core microbiomes with similar functions. Researchers have hypothesized that increased microbial diversity in the gut confers resilience, thereby promoting health and preventing disease [44]. The term “species richness” is often used to describe the number of species present in someone’s microbiome, and those with more species-rich microbial communities are less susceptible to invasion by pathogenic microbes. 

The importance of microbiome diversity has recently been highlighted in research examining the associations between skin and gut microbiota and dermatologic conditions such as atopic dermatitis, acne, rosacea, and psoriasis. In general, it appears that increased diversity of the entire microbiome plays a beneficial role in promoting gut health. With skin, it is more complex. While skin microbiome diversity is important, the role of sebum can be influential to controlling diversity. In sebum-rich areas lipophilic bacteria are more dominant and drier skin may promote a more diverse microbiome. However, because increases in diversity can correlate with increased skin dryness, this may not be healthy if it implicates an impaired skin barrier. 

## 4. Disease-Specific Changes

### 4.1. Acne

The acne microbiome has been studied for several decades, beginning first with culture based work in the 1960s and continuing now with sequencing surveys [1,45,46,47,48]. Perhaps in part due to its taxonomy, *Cutibacterium acnes* (formerly *Propionibacterium acnes*) was the presumed causative agent of acne vulgaris, though there is still a lack of consensus. There are at least three well-recognized sub-groups of *C. acnes* (I, II, and III), but up to nine ribotypes have been parsed [45,49,50,51,52,53]. The acne-associated *C. acnes* strains differ mainly in (putative) lipases, carbohydrate transport and metabolism, putative antimicrobial peptides, and putative virulence factors (CAMP and *tly* homologues) [54,55]. Barnard et al. also hinted that *C. acnes* bacteriophages may play a role in pathology, with phage being more prevalent and in higher abundance in control subjects, presumably controlling the abundance of *C. acnes* within the follicle [56,57]. 

While *S. epidermidis* has long been known as an opportunistic pathogen of soft tissues [58], its potential role in acne pathology has been largely ignored. This is despite several studies which show it is more abundant on acne subjects and on active lesions [59,60,61]. However, one study suggests that *S. epidermidis* may support recovery in acne lesions [62], though there appear to be some methodological shortfalls. 

*Malassezia* species, and more recently *Candida* species, have also been implicated in acne pathology, causing well-documented cases of folliculitis [63,64,65,66,67,68]. However, researchers are usually careful not to call yeast-associated folliculitis “acne”, due to the perception that acne vulgaris includes other clinical findings such as comedones that are not typically found in yeast-associated folliculitis. 

The causes of acne are complex and are promoted by an intermicrobial interaction rather than by the mere presence of a particular microbe. Some have attempted to harness the resident microbiome to create an acne therapy by decreasing *C. acnes* colonization [69,70], but none of these technologies have successfully made it to market or proven efficacy in vivo.

There has been a distinct paucity of studies examining the microbiome before and after successful treatments, with most using healthy controls and acne subjects. Given that the mechanisms of the most prescribed acne medications, including antibiotics, are largely unknown, this could be a fruitful approach to understanding the skin microbiome and the etiology of acne.

### 4.2. Atopic Dermatitis

Atopic dermatitis (AD), often colloquially referred to as eczema, is a chronic allergic skin disease characterized by an erythematous, dry, and intensely pruritic rash in a distinctive distribution [71]. The disease burden of AD is high, as it afflicts up to 20% of infants and 3% of adults worldwide and is often associated with other diseases like allergic rhinitis and asthma [72,73]. The cause of AD is known to be multifactorial, with both genetic and epigenetic factors contributing [71]. Notably, the incidence of AD is much higher in industrialized countries, and has continued to increase for decades, suggesting that excessive hygiene may be affecting the body’s microbial milieu and inhibiting the body from adapting beneficial immune responses to pathogens [74,75,76,77]. Numerous studies have explored the role of the skin microbiome and its effects on the clinical manifestations of AD.

In AD, gene defects lead to both physical and Th2-mediated immunological disruptions in the skin barrier, precipitating increased susceptibility to infection and allergens [78]. This barrier dysfunction is then exacerbated by the physical stress of repeated scratching of dry, itchy skin [79]. Th2 cytokines can suppress keratinocyte induction of antimicrobial peptides (AMP) such as human beta-defensin-3 and cathelicidins, both of which prohibit colonization of pathogenic organisms such as *S. aureus*, thereby maintaining microbiotic homeostasis [80,81]. Over time, the skin microbial flora become altered from their usual composition, driving the skin towards a diseased state. Further contributing to our understanding that certain microbial communities play a crucial role in the manifestation of this disease, AD classically involves the antecubital and popliteal fossae—body sites that host similar compositions of specific organisms [82]. 

*Staphylococcus aureus* is consistently implicated in AD, and has been shown to colonize lesional AD skin at high rates—much more so than nonlesional atopic skin or the skin of healthy subjects [83,84,85,86]. Furthermore, a higher density of colonization with *S. aureus* is correlated with more inflammation and increased disease severity [84,87,88]. However, it is not just the types of bacteria, but also the diversity of organisms, that play a role in AD, as a lack of cutaneous microbial diversity contributes to disease pathogenesis. A study by Kong et al. demonstrated that in lesional AD skin the microbiome is dominated by *staphylococcus* species, particularly *S. aureus*, thereby decreasing the overall diversity of other microorganisms [4]. Another clinical investigation showed that infants colonized with *S. epidermidis* and *S. cohnii* by two months of age had a significantly reduced risk of developing eczema by one year of age, possibly due to increased microbiome diversity [89]. A 2017 study by Nakatsuji et al. demonstrated that compared to the skin of AD subjects, healthy subjects had much higher levels of coagulase-negative *Staphylococcus* species (specific strains of *S. epidermidis* and *S. hominis*) with anti-*S. aureus* activity. When isolated and introduced into human subjects with AD, these coagulase negative strains decreased cutaneous *S. aureus* colonization [10]. Supporting these findings, a recent pediatric trial found that patients with mild AD flares had more *S. epidermidis* detected, while those with severe disease were colonized by dominant *S. aureus* strains [90]. These discoveries illustrate that a properly functioning skin microbiome protects against pathogenic organisms, whereas disruption in the microbiotic environment encourages atopic disease.

Though atopic dermatitis is considered a skin disorder, the gut microbiome is also thought to play a role in disease pathogenesis. Many investigations exploring this relationship are performed in infants and children, as this is the demographic with the highest disease prevalence [71]. Similar to the skin microbiome, both the presence of specific organisms and the degree of microbiotic diversity are implicated. Several prospective studies show that the colonic microbiomes of infants who develop AD later in life have lower levels of microbial diversity [91,92,93,94]. These prospective studies highlight that irregularities in gut microbiota precede manifestations of AD and may warrant further study to assess if there is a causative role. 

Researchers have also explored the prevalence of various types of gut bacteria in patients with and without eczema. As in the skin, studies have noted higher counts of *S. aureus* in fecal samples of AD subjects [95,96]. Delving deeper, a recent publication found that the *S. aureus* strains in infants (0–2 months of age) who developed AD by 18 months of age were less likely to have genes encoding for a specific superantigen and elastin-binding protein [97]. It was postulated that these bacterial proteins help the infant immune system mature, thus preventing atopic disease. Aside from *S. aureus*, many other bacteria species have been found to be more prevalent in AD individuals. Both prospective and cross-sectional studies have found higher prevalence of clostridia species among children and adults with AD [95,96,98]. In a Dutch cohort, colonization with *Clostridium difficile* in the first month of life was associated with AD up to age seven, and colonization with *Escherichia coli* was associated with a higher risk of AD at age two [99,100]. In addition, studies of both adult and infant populations have observed greater colonization with enterobacteriaceae in AD [101,102]. *Faecalibacterium prausnitzii* has also been found to be significantly elevated in the AD population [103,104]. Other bacteria are found to less frequently colonize AD populations, such as bifidobacteria [95,102]. Although there is still much work to be done in identifying and analyzing gut microbiota in AD compared to healthy populations, the science shows that there are several distinct differences between these groups that are likely contributing to disease manifestation.

Patients with AD are typically treated with a combination of emollients and antimicrobial, anti-inflammatory, and/or systemic immunosuppressant medications. More recently, researchers have explored how these interventions affect not only disease severity, but also the composition of the skin microbiome. Open and double-blind placebo-controlled trials have demonstrated that oral and topical antimicrobial treatments can reduce *S. aureus* skin colonization and lead to improvement in disease severity [83,105,106]. However, studies exploring the effects of systemic antibiotics alone have not demonstrated long-term improvement in AD skin lesions and revealed only transient decreases in the cutaneous *S. aureus* burden [107,108]. A study comparing the efficacy of a combination topical steroid/topical antibiotic versus steroid alone also failed to show superior improvement in AD severity or *S. aureus* colonization in the antibiotic-treated cohort [85]. More than just decreasing *S. aureus* colonization, studies also clearly show that treatment of AD lesions with a variety of interventions restores diversity to the cutaneous microbiota. A single-blind placebo-controlled trial demonstrated that treatment with a topical corticosteroid alone was non-inferior to steroids plus bleach baths in improving AD lesions and restoring the microbial milieu to that of non-lesional AD skin [109]. Similarly, a study investigating microbial populations on intermittently-treated AD skin (topical corticosteroids, calcineurin inhibitors, or antibiotics in the last seven days, and/or oral antibiotics within the last four weeks) versus untreated AD skin, found that intermittently-treated skin had less *S. aureus* and more microbial diversity [4]. Additionally, increases in *Streptococcus*, *Corynebacteria*, and *Propionibacteria* were seen following therapy. Another study found that even when only emollients are used in lesional AD skin, the diversity of cutaneous microorganisms is increased and restored to that of unaffected skin [110]. Prevention and treatment of AD through the administration of oral probiotic supplements in pregnancy and in infancy has shown modest improvement in AD in some cases, but has yet to become a mainstay of treatment therapy, and the effects of supplementation on the microbiome specifically are less explored [111]. These data emphasize the value of modulating the dysbiotic microbiome, through a variety of interventions, as an important therapeutic principle of AD treatment.

While it is tempting to draw conclusions from the expanding literature on the microbiome and atopic dermatitis, there are several caveats that must be kept in mind. Studies on the topic involve populations from different countries, with unique diets, hygiene, and genetic predispositions that all have the potential to alter or confound findings. Moreover, as technology has advanced, more detailed information about the microbiotic composition has become available, and we must ask whether undetected bacterial strains were present in the subjects of earlier studies. Lastly, there are sites along the aerodigestive tract outside of the intestines that may contribute to the microbiome and play an additional part in the development of AD. In order to identify how the microbiomes of each body site may work together to influence the manifestation of AD, large studies among well-characterized birth cohorts with long-term follow-up are essential. Understanding how all of the bacterial communities interact and affect one another is needed to fully understand and effectively treat this disease. 

### 4.3. Rosacea

Rosacea is a chronic inflammatory condition of facial skin that affects between 0.9% and 10% of the American and European population [112]. It is classically identified by facial flushing or persistent facial erythema, telangiectasia, and/or inflammatory papules and pustules [113]. The pathogenesis is not fully understood but the clinical manifestations of rosacea are multifactorial and are at least in part due to abnormal neurovascular activation, dysregulated production and release of inflammatory molecules, and overgrowth of organisms that naturally inhabit the skin [114]. 

How the cutaneous microbiome differs in rosacea has been the subject of many scientific investigations. *Demodex folliculorum* (a.k.a. demodex), a mite that lives in the sebaceous glands of healthy skin, is a commonly implicated pathogen in rosacea, as it has been found in numerous studies to exist in excess on the skin of afflicted patients [115,116,117,118]. Further, skin biopsies with higher counts of demodex show prominent inflammatory cell populations around hair follicles and have greater expression of genes encoding inflammatory peptides and cellular growth factors [117,119]. It has been postulated that the mite exoskeleton stimulates production of some of these pathogenic mediators [120]. Still, demodex is unlikely to be the only cutaneous microbiotic component contributing to the disease. One study showed that although treatment of rosacea with topical anti-demodex cream (permethrin 5%) decreased demodex counts significantly, it was not superior to topical antibiotics (metronidazole 0.75%) in improving rosacea, suggesting bacterial pathogens may be involved [121]. Demodex mites are suspected carriers of *Bacillus oleronius*, pro-inflammatory, gram-negative bacteria that are susceptible to many antibiotics commonly used to treat rosacea, including doxycycline [122,123,124]. *Staphylococcus epidermidis*, a healthy skin commensal, has also been isolated from rosacea pustules [125]. Yet, in comparison to the non-hemolytic *S. epidermidis* found on healthy controls, the *S. epidermidis* isolated from rosacea patients was a beta-hemolytic variant with possible increased virulence [126]. It was previously presumed that antibiotic therapy improved rosacea primarily through its anti-inflammatory effects, but these studies suggest that an anti-bacterial mechanism is also at play.

Alterations in the gut microbiome have also been implicated in rosacea pathogenesis. A small cross-sectional study comparing the gut microbiota of those with and without rosacea found several differences between groups, with certain bacteria being more abundant in rosacea patients, and other bacteria populations being less abundant. In contrast with atopic dermatitis, metagenomics showed no difference in the microbiotic diversity of subject groups [127]. The most-studied gut bacterium in its relation to rosacea is *Helicobacter pylori*, a gram-negative organism that resides in the stomach of approximately 50% of the population [128,129,130,131]. Although *H. pylori* colonization is usually asymptomatic, it is also a well-known cause of dyspepsia, gastric and duodenal ulcers, and various GI malignancies [132]. Assessing *H. pylori*’s contribution to rosacea can be difficult, as standard treatment of both afflictions is with oral antibiotics. However, in studies that controlled for prior antibiotic use, *H. pylori* IgG seropositivity was strongly correlated with rosacea [133,134]. Further supporting an association, rosacea severity has been shown to decrease with eradication of *H. pylori* [135,136,137]. Again, it is unclear if treatment with antibiotics confounded these results. Though the exact pathway between *H. pylori* infection and rosacea has not been fully elucidated, studies suggest it may exert effects through its pro-inflammatory virulence peptides, particularly in those with concomitant gastrointestinal symptoms [138,139]. Nonetheless, the association with *H. pylori* and rosacea remains controversial, as other studies have not found a correlation between the two entities [140,141,142]. It is still debated whether dysbiosis occurs in response to rosacea, or is a cause [114]. Ultimately, more studies exploring the microbiome and rosacea are needed to further clarify their relationship with one another.

### 4.4. Psoriasis

Psoriasis is a chronic inflammatory skin disease that is estimated to affect 2–3% of the population, often appearing between 15–25 years of age [143]. Classically, it is characterized by raised, scaly, well-demarcated, erythematous lesions on extensor surfaces and can be intensely pruritic [143]. In some cases, extracutaneous manifestations of psoriasis, such as ocular or joint involvement can be seen [144]. Although the mechanism of disease is unclear, its pathology is believed to be multifactorial, involving strong genetic factors, disruptions to the immune system, and environmental triggers [145]. 

Investigation into the skin microbiome has cemented the role of skin flora dysbiosis in psoriasis. Many have attempted to characterize the bacterial composition of the skin microbiota in psoriatic lesions. Gao et al. used 16S rDNA PCR to compare affected to non-affected skin from 6 patients and found that *Firmicutes* was significantly overrepresented in psoriatic lesions compared to healthy skin, while *Actinobacteria* was significantly underrepresented [146]. However, another study found the complete opposite—there was a significant decrease in *Firmicutes* and *Staphylococcus* in affected skin compared to non-affected skin [147]. A similar study identified *Corynebacterium*, *Propionibacterium*, *Staphylococcus*, and *Streptococcus* as the major bacterial genera located on lesional and non-lesional skin from psoriasis patients, with a significant abundance in *Firmicutes* and *Actinobacteria*. There was a continual decrease in bacterial diversity observed in non-lesional and lesional skin from psoriasis patients when compared to healthy controls, suggesting that dysregulation of the skin microbiome seen in psoriasis is not limited to lesional skin, but affects the entire cutaneous microbiota as a whole [148]. 

Interestingly, *S. aureus* is known colonize the skin in psoriasis more abundantly in comparison to those without psoriasis [149], although it is rare for *S. aureus* to cause an overt infection. Chang et al. also found an abundance of *Staphylococcus aureus* in lesional and non-lesional skin from patients with psoriasis, compared to healthy controls [150]. To understand the significance, they used a mouse model to explore the effects of *S. aureus* on T cell differentiation and found that mice colonized with *S. aureus* demonstrated a strong Th17 polarization compared to mice colonized with *S. epidermis* [150]. Their results suggest that *S. aureus* is capable of upregulating a Th17 response, initiating the release of pro-inflammatory cytokines, and contributing to the inflammation observed in psoriasis. These cumulative efforts have led to the conclusion that psoriasis is associated with dysregulation in composition and a decrease in diversity of the cutaneous microbiota at both affected and unaffected sites. However, characteristics of the bacterial microbiome in a disease versus healthy state remains unclear.

Few studies have looked at the effect of treatment on the skin microbiome in psoriasis. One study analyzed the skin microbiome of patients with chronic plaque-type psoriasis before and after treatment with narrowband UVB (nbUVB). Those with at least a 75% reduction in the Psoriasis Severity Index (PSI) score demonstrated a significant decrease in *Firmicutes*, *Pseudomonas*, *Staphylococcus*, *Finegoldia*, *Anaerococcus*, *Peptoniphilus*, *Gardnerella*, *Prevotella* and *Clostridium* after nbUVB treatment [147]. Another study found that 3 weeks of selenium-rich water balneotherapy treatment resulted in an increase in *Xanthomonadaceae*—a bacteria with known keratolytic activity—and subsequent clinical improvement in the Psoriasis Area and Severity Index (PASI) score [151]. These limited studies suggest a promising role for microbiome-targeted treatment, but a causal relationship between successful treatment and modification of the skin microbiome cannot be concluded based on these results alone. 

Psoriasis, like many other systemic inflammatory diseases, likely involves inappropriate activation of various immune pathways leading to elevations in pro-inflammatory cytokines. The gut microbiome is believed to be involved in the development of pro-inflammatory Th17 cells, allowing it to modulate inflammation in diseases such as inflammatory bowel disease and obesity [152]. Like the skin microbiota, the composition of the gut microbiota and its relation to psoriatic disease is unclear. Tan et al. compared the gut microbiota in patients with psoriasis to those without and found that those with psoriasis had a significant decrease in *Akkermansia muciniphila*, a species believed to strengthen the integrity of the gut epithelium and protect against systemic inflammatory diseases such as inflammatory bowel disease, obesity, and atherosclerosis [152,153,154,155,156]. Scher et al. found a decrease in bacterial diversity in the gut of patients with psoriatic arthritis and skin-limited psoriasis [157]. Two groups described a decrease in *Actinobacteria* compared to healthy controls. One group found that an elevated *Firmicutes* to *Bacteroidetes* ratio in psoriatic patients, was positively correlated with PASI score [157,158]. Interestingly, this abnormality is also correlated with the increased inflammation observed [159,160]. However, contrary to these some of these findings, Codoñer et al. saw an increase in bacterial diversity, overrepresentation of *Faecalibacterium*, *Akkermansia*, and *Ruminococcus*, and a decrease in *Bacteroides* in 52 patients with psoriasis [161]. Several confounding factors could play a role in these discrepancies including differences in study population as well as changes in technology. 

Although most treatment for psoriasis do not involve direct modulation of the gut microbiome, evidence suggests that there may be a potential role for it. One study explored the role of the gut microbiota in murine-models of psoriasis. They found that mice reared in germ-free conditions and mice treated with antibiotics developed less severe imiquimod-induced skin inflammation than conventional mice. Mice treated with antibiotics had a significant increase in *Lactobacilalles*—lactic-acid producing bacteria with anti-inflammatory effects in a healthy gut. The mice reared in germ-free conditions and mice treated with antibiotics also had a reduction in Th17 cells compared to conventional mice [162]. Together, these findings demonstrate that a lack of gut microbiota results in resistance to the development of psoriasis, possibly through preventing the development of Th17 cells and a reduction in the pro-inflammatory pathway. This study further elucidates the importance of the gut microbiome in the development of psoriasis and suggests a role for antibiotics in reducing acute exacerbations of psoriasis. Future studies are needed to see if these findings can be extrapolated to human subjects. 

### 4.5. Seborrheic Dermatitis 

Seborrheic dermatitis (SD) is a common inflammatory rash that most often occurs on areas of skin with a high density of sebaceous glands, such as the scalp, face, and trunk [163]. Outbreaks of SD often correlate with specific triggers, such as weather change, depression, and emotional stress [164]. The incidence of SD peaks at three ages: infancy, teenage years, and adults over 50 years old, suggesting that hormonal changes in sebum production play one role in its pathogenesis. Although SD commonly occurs in generally healthy individuals, it is especially prevalent in people with Parkinson’s disease or those infected with human immunodeficiency virus (HIV) [165]. In addition to being associated with sebum-rich hair-bearing areas, seborrheic dermatitis is associated with *Malassezia*, ubiquitous fungi that is normally a part of the human skin microbiome. However, the role of *Malassezia* in the development of SD is still poorly understood [166]. Tanaka et al. recently used pyrosequencing and quantitative polymerase chain reaction to characterize the bacterial microbes on skin with and without SD. The investigators found that *Actinetobacter*, *Staphylococcus*, and *Streptococcus* dominated the skin microbiome of lesional skin affected with SD compared to nonlesional skin [167]. In a cohort study of patients with SD, An et al. found that patients with SD had a significantly higher colonization with *Staphylococcus epidermidis* than those without SD [163]. To better understand this association, more clinical studies are needed investigating how the skin microbiome compares between affected and unaffected individuals, and how the microbial composition is changed after treatment. As researchers learn more about the role of the skin microbiome in the etiology of SD, more targeted treatment approaches can be developed towards implicated microbial organisms. 

## 5. Role of Probiotics and Prebiotics 

Probiotics are defined as “live microorganisms which when administered in adequate amounts confer a health benefit to the host” [168]. Probiotics have become more popular over the last twenty years, as research has suggested that they may be beneficial to many aspects of human health such as in the prevention of antibiotic-associated diarrhea and in the treatment and prevention of infectious diarrhea [169]. The most commonly used bacteria are lactobacilli and bifidobacteria [170]. Prebiotics are non-digestible food ingredients, which are able to benefit the host by selectively stimulating the growth and or activity of bacterial species present in the colon. The most common prebiotics are non-digestible oligosaccharides. Synbiotics are a combination of prebiotics and probiotics [171].

There have been numerous studies investigating the use of prebiotics and probiotics in both the treatment and prevention of atopic dermatitis (AD) in children. A recent meta-analysis by Huang et al. identified thirteen studies in children with AD. They found increased efficacy of probiotics in treating AD in children in the 1–18 year subgroup [172]. Regarding disease prevention, a meta-analysis by Lee et al. found a significant risk reduction in pediatric atopic dermatitis with the administration of prenatal or postnatal probiotic supplementation. However, they did not find evidence to support the use of probiotics in the treatment of established AD [173]. A reduction in pediatric atopic dermatitis following maternal supplementation has also been supported by other meta-analyses [174]. In contrast, meta-analyses have generally failed to demonstrate much benefit of probiotics in the treatment of established AD. However, it should be noted that individual trials assessing treatment outcomes were smaller and exhibited more heterogeneity in comparison to the prevention studies [173]. Far fewer studies have been conducted addressing the use of probiotics in the treatment adults with AD. A meta-analyses of four studies in adults with AD found a mean weighted difference of –8.26, 95% CI –13.28 to –3.25 [173].

There is one reported case of probiotics being used to successfully treat pustular psoriasis in India [175]. However, a randomized control trial examined the use of probiotics in the treatment of spondyloarthritis patients, including psoriasis and psoriatic arthritis, and found no significant change in disease severity compared to placebo [176]. Further research needs to be conducted regarding the use of probiotics in other dermatological disease such as acne, rosacea, wound healing, and psoriasis [177]. Probiotics should not all be classified as one intervention as there are differences among probiotics depending on the species, genus, or where they are a spore-former or not. 

## 6. On the Horizon: Gut-Skin Communication

If it is hypothesized that changes to the gut microbiome can lead to cutaneous manifestations, the question becomes, ‘how?’. The “brain-gut-skin axis”, was initially proposed by John H. Stokes and Donald M. Pillsbury in 1930 [178]. They hypothesized that negative emotional states such as depression and anxiety alter the gastrointestinal function and lead to changes in normal gut flora, increased intestinal permeability, and systemic inflammation [178]. A study demonstrating that mice fed probiotics demonstrated reduced stress-induced neurogenic skin inflammation when compared to control mice. Although the mechanism was not elucidated, the authors suggested that increased epithelial permeability in the gut triggers T-cell activation and disrupts immunosuppressive cytokines and T regulatory cells responsible for establishing tolerance, leading to systemic inflammation that can disrupt cutaneous homeostasis [179]. Additionally, it has been suggested that altered levels of neurotransmitters, such as acetylcholine, norepinephrine, and dopamine produced by gut organisms can communicate with peripheral organs through neuronal pathways not yet identified [180].

Another theory is that increased intestinal permeability associated with altered gut flora allows for direct migration of inflammatory products into the systemic circulation. In a study involving obese mice, Cani et al. demonstrated that modifications in diet, namely excess dietary fat, facilitates increased absorption of highly proinflammatory lipopolysaccharide (LPS, a membrane component of Gram-negative bacteria) from the gut, and contributed to chronic low-grade systemic inflammation and the development of metabolic disease [181]. Research in psoriasis patients found that they actually had increased levels of bacterial DNA in the bloodstream compared to healthy controls. In a study involving 54 patients with psoriasis and 27 controls, bacterial DNA was detected in the bloodstream of 16 patients with plaque psoriasis and in none of the control patients [182]. These patients with serum bacterial DNA also exhibited significantly higher levels of systemic inflammatory response markers, including IL-1β, IL-6, IL-12, tumor necrosis factor, and interferon γ. It is presumed that the bacterial DNA would have originated from the intestinal lumen and suggests that decreased integrity of the gut epithelium is involved in the pathogenesis of psoriasis. A 2018 study did not identify a specific bacterium responsible for bacterial DNA translocation (BT), arguing that there is not a specific gut microbial composition that allows for increased likelihood of bacterial DNA translocation, but rather, inflammation from an overall microbial imbalance [161]. 

Although it is apparent that there is a link between the gut microbiota and dermatological disease, the exact mechanism is poorly understood. Current evidence suggests that it is likely due to a combination of both neurologic and immunologic responses to environmental shifts, resulting in chronic systemic inflammation that can ultimately affect the skin. Further studies need to go beyond descriptive microbiome studies, to include functional characteristics such as transcriptomics, lipidomics, and secondary metabolite measures, as this will allow us to better understand the method of communication between the gut and the skin in order to design therapies that effectively target multiple aspects of disease pathogenesis.

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
