# Peer review of "The Skin and Gut Microbiome and Its Role in Common Dermatologic Conditions"

_microorganisms, 2019, doi:10.3390/microorganisms7110550_

Round 1

Reviewer 1 Report

The review manuscript by Dr. Ellis et al., describes potential roles of the microbiome on development of skin disorders. The manuscript covers a wide range of information regarding correlation between very common skin disorders and dysbiosis of microbial communities. Therefore, this review manuscript would be of a great interest to readers of microorganisms. I only have minor comments described below.

1) I recommend discussing how the skin microbiome, especially staphylococcus species, contributes to the cutaneous homeostasis and development of cutaneous immune systems, under the “The Skin Microbiome” section.  A lot of current publications have approached these mechanisms.

2) It has been well known that Th2 cytokines suppress induction of antimicrobial peptides, such as Cathelicidin and beta-defensins, a mechanism which contributes to development of dysbiosis in atopic dermatitis.  This must be discussed.

3)  As discussed in the manuscript, development of Th17 immune response is one abnormal characteristic in psoriatic skin.  It has been well known that excessive production of antimicrobial peptide, LL-37, is greatly involved in the development of Th17 immune response. On the contrary, the authors describe possible contribution of S. aureus in the Th17 immune response in psoriasis in lines 313-322.  However, skin colonization by S. aureus is rarely found in psoriasis.  It is true that S. aureus is a key regulator of Th17 in wound infection, but it may not be the case in psoriasis.  The description may mislead readers.  Authors must carefully discuss these topics.   

Reviewer 2 Report

The manuscript (MS) “The Skin and Gut Microbiome and Its Role in Common Dermatologic Conditions”, submitted by Ellis et al., reviews a great number of articles (179 references) in this field. The skin conditions discussed in this review are acne, AD, SD, rosacea and psoriasis. At the end, authors briefly discuss the potential application of probiotics and/or prebiotics on therapeutic regimens for these diseases and postulate a gut-skin communication mechanism.  

The MS is quite well written, with enough evidence either to support or deny -- as appropriate -- any of the specific species- or flora-related skin conditions.  However, the results cited from the references seem to be focused mainly on the descriptive observation, and the MS's purview exhibits an alarming lack of mechanism studies, which this reviewer believes will limit its novelty for its intended audience.
